# Diversity and Ecology of *Lobophora* Species Associated with Coral Reef Systems in the Western Gulf of Thailand, including the Description of Two New Species

**DOI:** 10.3390/plants11233349

**Published:** 2022-12-02

**Authors:** Anirut Klomjit, Christophe Vieira, Felipe M. G. Mattos, Makamas Sutthacheep, Suttikarn Sutti, Myung-Sook Kim, Thamasak Yeemin

**Affiliations:** 1Marine Biodiversity Research Group, Department of Biology, Faculty of Science, Ramkhamheang University, Bangkok 10240, Thailand; 2Research Institute of Basic Science, Jeju National University, Jeju 63243, Republic of Korea; 3Taiwan International Graduate Program, Biodiversity Research Center, Academia Sinica, Taipei 115, Taiwan; 4Thailand Natural History Museum, National Science Museum, Pathum Thani 12120, Thailand

**Keywords:** *cox*3, diversity, *Lobophora*, macroalgal–coral interaction, *psb*A, *rbc*L

## Abstract

The brown macroalgal genus *Lobophora* plays important ecological roles in many marine ecosystems. This group has received much attention over the past decade, and a considerable number of new species have been identified globally. However, our knowledge of the genus diversity and ecology along south-east Asian coasts are still limited. Given the growing body of research that uses a combination of molecular and morphological data to identify cryptic species, this study investigates the diversity of *Lobophora* in the western Gulf of Thailand using morphological and molecular data, as well as their interactions with scleractinian corals. A total of 36 *Lobophora* specimens were collected from 15 sites in the western Gulf of Thailand and used for molecular and morphological analyses. One mitochondrial (*cox*3) and two chloroplast (*psb*A and *rbc*L) genes were amplified and sequenced for molecular phylogenetic analyses. Based primarily on phylogenetic evidence, two new species were formally described, *L. chumphonensis* sp. nov. and *L. thailandensis* sp. nov. Additionally, *L. lamourouxii* was newly recorded from Thailand. Two new lineages of *Lobophora obscura* were identified, *L. obscura*12 and *L. obscura*13. Among the *Lobophora* species identified, three were found in interaction with corals, the most notable of which was the massive coral *Porites*. *Lobophora chumphonensis* sp. nov. only interacted with *Porites* by growing on bare coral skeleton between *Porites* colonies. Furthermore, *L. obscura*13 was observed under the branching coral *Pocillopora*. Our findings revealed that *Lobophora* presented both effects and absence of effects on coral. A thorough understanding of *Lobophora* diversity and ecology is essential for ongoing and future research on coral–macroalgal ecological relationships.

## 1. Introduction

The brown algal genus *Lobophora* J. Agardh is naturally associated with coral reefs in tropical and subtropical seas [1]. *Lobophora* species are characterized by a relatively small thalli with a simple morphology, ranging from erect to crustose morphotypes [1]. A more detailed perspective of the species diversity and distribution of *Lobophora* has been obtained in recent years [2,3,4,5,6,7,8]. Currently, a total of 71 species are taxonomically accepted according to the online database for algae, AlgaeBase [9]. Moreover, the diversity of *Lobophora* was estimated to surpass 100 species worldwide [5]. 

*Lobophora*–coral interactions have been reported across global coral reef systems, including but not limited to the South Pacific, such as Great Barrier Reef and New Caledonia [10,11]; the Caribbean Sea, such as Belize, the Bahamas, Curaçao, and the Mesoamerican Barrier Reef [12,13,14,15]; and the Central Indo-Pacific, such as Palau, Okinawa, and the Malacca Strait [16,17,18]. The brown alga *Lobophora* plays an ecologically significant role in coral ecosystems as food resources for coral reef herbivores and a habitat for marine organisms [19,20,21,22]. However, negative effects of *Lobophora* species toward corals have also been reported in several ways, e.g., overgrowth and killing healthy corals, and allelopathic effects on coral recruitment, tissue (i.e., bleaching, diseases), and bacterial communities (i.e., community alteration) [10,23,24,25,26,27,28].

Prior to the use of molecular tools for this genus, *L. variegata* has been widely reported in Thailand, including in the Gulf of Thailand [29,30,31,32]. More recent studies that have made use of molecular tools revealed the presence of three putative species of *Lobophora* (*L.* sp.24, *L.* sp.49, and *L.* sp.56) in the Southern Andaman Sea, Thailand [4,33], but not the genuine *L. variegata*. A recent study on the genus diversity from the Western Indian Ocean unveiled an important diversity in this region, with no less than 43 species, which stands in contrast with the only 3 species recorded to date in the Gulf of Thailand adjacent to the Indian Ocean [34]. There is a clear gap in knowledge in this part of the world for the diversity of the genus *Lobophora* species. Comparatively, our knowledge of *Lobophora* ecology in this region is very limited. No ecological studies have been conducted in Thailand, including on *Lobophora*–coral interactions. This study aimed to document *Lobophora* species diversity and ecology in coral reef areas in the western Gulf of Thailand and interactions with scleractinian corals on coral reef systems in the western Gulf of Thailand.

## 2. Results

### 2.1. Taxonomic Results

A total of 36 algal specimens were successfully sequenced with *cox*3, and 16 algal genomic DNA from representatives of each species were sequenced with *psb*A and *rcb*L genes. The molecular analyses based on the individual and concatenated alignments of *cox*3, *rbc*L, and *psb*A sequences revealed five/six *Lobophora* lineages from coral reefs in the western Gulf of Thailand (Figure 1), including one previously described species, *L. lamourouxii*, Payri and C.W. Vieira in [8]; one previously identified lineage part of the *L. pachyventera* complex, *L. pachyventera*4 from Malaysia (Appendix A); one lineage related to *L. abscondita*, C.W.Vieira, Payri, and De Clerck, and *L.* sp82; and finally two to three lineages part of the *L. obscura* complex. Based on *cox*3 and concatenated trees, the *L. obscura* lineages presently identified from Thailand are closely related to *L. obscura*7, *L. obscura*9 and *L. astrolabeae*, C.W.Vieira and Payri [8], whereas resolution of relationships among these species is less clear in the chloroplast trees (*psb*A and *rbc*L) (Appendix A). Based on molecular and morphological data, the lineage related to *L. abscondita* and the one corresponding to *L. pachyventera*4 are herein formally described as two new species: *Lobophora chumphonensis* sp. nov. and *Lobophora thailandensis* sp. nov. 

#### 2.1.1. *Lobophora chumphonensis* sp. nov. A.M.Klomjit and C.W.Vieira (Figure 2g,h, Table 1)

Description: Large orbicular thallus, up to 40 cm in diameter, thin, strongly adherent to the substratum across the whole of the ventral surface by abundant rhizoids, dark brown in color. Thallus composed of single-cell-layered medulla, two to six- and two to four-cell-layered cortex on the dorsal and ventral side, respectively. The thallus was 75–105 μm thick and composed of 5–11-cell layers. The sporangium was 36–48 μm in height and 60–78 μm in width.

**Figure 2 plants-11-03349-f002:**
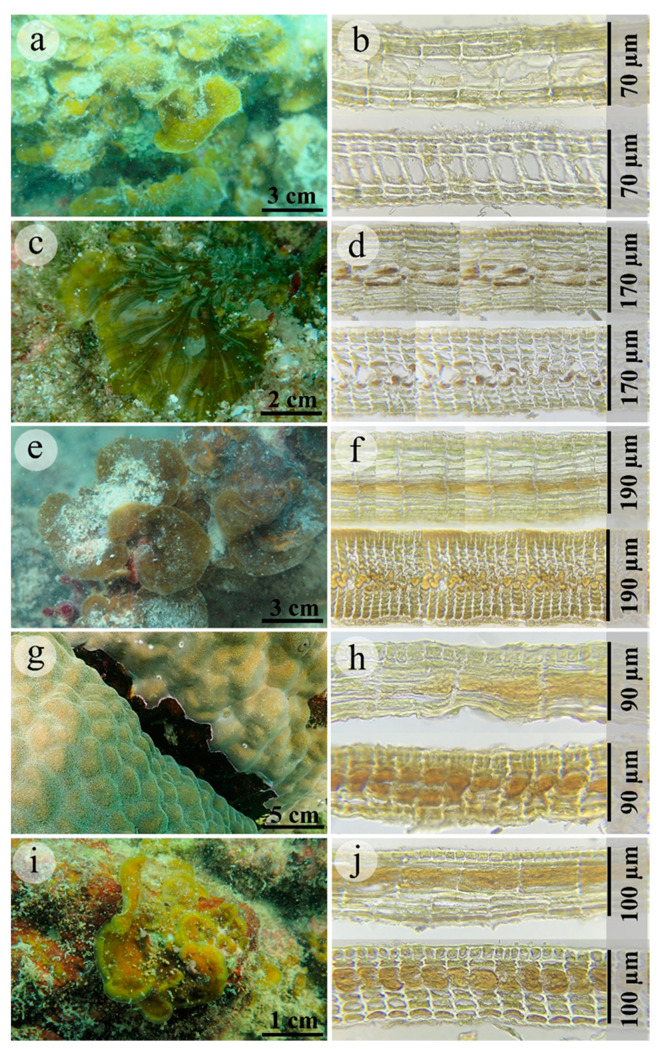
In situ photographs and corresponding transverse and cross-section through middle of thallus: *Lobophora lamourouxii*, THNHM-P-2020-0291 (**a**,**b**), *Lobophora obscura*13, THNHM-P-2020-0262 (**c**,**d**), *L.obophora obcura*12, THNHM-P-2020-0277 (**e**,**f**), *Lobophora chumphonensis* sp. nov., THNHM-P-2020-0258 (**g**,**h**), *Lobophora thailandensis* sp.nov., THNHM-P-2020-0287 (**i**,**j**).

Type locality: Chumphon archipelago, Ang Thong archipelago.

Distribution: Thailand.

Holotype: THNHM-P-2020-0258, collected 5 March 2020, deposited in the Herbarium of Natural History Museum of the National Science Museum in Thailand.

Habitat: This species is commonly grown on hard substratum between free-space *Porites* colonies or grown on the upper dead space of *Porites* colonies in various habitats including fringing reefs and shallow reef flats.

Etymology: the specific epithet refers to Chumphon, which is the locality where the materials were collected. 

Specimens: Mattra Island, Chumphon, Thailand, 5 March 2020, *leg*. A.Klomjit (THNHM-P-2020-0258); Reat Island, Chumphon, Thailand, 5 March 2020, *leg*. A.Klomjit (THNHM-P-2020-0257); Sam Sao Island, Surat Thani, Thailand, 24 March 2020, *leg*. A.Klomjit (THNHM-P-2020-0259); Sam Sao Island, Surat Thani, Thailand, 24 March 2020, *leg*. A.Klomjit (THNHM-P-2020-0260); Sam Sao Island, Surat Thani, Thailand, 24 March 2020, *leg*. A.Klomjit (THNHM-P-2020-0261). 

#### 2.1.2. *Lobophora thailandensis* sp. nov. A.M.Klomjit and C.W.Vieira (Figure 2i,j, Table 1)

[This species corresponds to *Lobophora pachyventera4* in Vieira et al. (2019)] 

Description: Flabellate thallus, up to 5 cm wide and 4 cm tall, thin, adherent to the substratum by rhizoids on the margin, dark brown to orange brown on dorsal in color, and orange brown to orange in color on ventral side. Thallus composed of single- to double-cell-layered medulla, two to seven- and two to four-cell-layered cortex on the dorsal and ventral side, respectively. The thallus was 87–110 μm thick and composed of 5–12-cell layers.

Type locality: Chumphon archipelago, Ang Thong archipelago, Samui Island, Taen Island, Pulau Redang.

Distribution: Thailand, Malaysia. 

Holotype: THNHM-P-2020-0287, collected 12 March 2020, deposited in the Herbarium of Natural History Museum of the National Science Museum in Thailand.

Habitat: This species is commonly grown on hard substratum, mostly dead corals and large rubbles, and can encrust on live corals in various habitats including fringing reefs and shallow reef flats.

Etymology: the specific epithet refers to Thailand, which is the locality where the materials were collected.

Specimens: Taen Island, Surat Thani, Thailand, 13 March 2020, *leg*. A.Klomjit (THNHM-P-2020-0286); Sam Sao Island, Surat Thani, Thailand, 24 March 2020, *leg*. A.Klomjit (THNHM-P-2020-0279); Sam Sao Island, Surat Thani, Thailand, 24 March 2020, *leg*. A.Klomjit (THNHM-P-2020-0280); Sam Sao Island, Surat Thani, Thailand, 24 March 2020, *leg*. A.Klomjit (THNHM-P-2020-0281); Sam Sao Island, Surat Thani, Thailand, 24 March 2020, *leg*. A.Klomjit (THNHM-P-2020-0282); Sam Sao Island, Surat Thani, Thailand, 24 March 2020, *leg*. A.Klomjit (THNHM-P-2020-0283); Tai Plao Island, Surat Thani, Thailand, 23 March 2020, *leg*. A.Klomjit (THNHM-P-2020-0284); Hin Dab Island, Surat Thani, Thailand, 23 March 2020, *leg*. A.Klomjit (THNHM-P-2020-0285); Samui Island, Surat Thani, Thailand, 12 March 2020, *leg*. A.Klomjit (THNHM-P-2020-0287); Samui Island, Surat Thani, Thailand, 12 March 2020, *leg*. A.Klomjit (THNHM-P-2020-0288); Kula Island, Chumphon, Thailand, 4 March 2020, *leg*. A.Klomjit (THNHM-P-2020-0289); Kula Island, Surat Thani, Thailand, 4 March 2020, *leg*. A.Klomjit (THNHM-P-2020-0290); Mak Kepit, Pulau Redang, Malaysia, 13 May 2008, *leg*. P.E.Lim (KU-d5184).

### 2.2. Morphological, Anatomical, and ECOLOGICAL Characteristics

*Lobophora obscura* exhibited two morphotypes corresponding to two to three different lineages that are part of the *Lobophora obscura* complex, one (hereafter identified as *L. obscura*12) closely related to *L. obscura*7 and the other one/two (hereafter identified as *L. obscura*13) to *L. obscura*9 and *L. astrolabeae*, based on the *cox*3 and concatenated trees. The anatomical characteristics of the *L. obscura* lineages, *L. chumphonensis* sp. nov., and *L. thailandensis* sp. nov. exhibited intraspecific variation, except that of *L. lamourouxii*, which exhibited identical characteristics within species in this study (Table 1 and Figure 2). A hierarchical clustering of *Lobophora* species by using morphology, thallus thickness, height, and width as predictors revealed that the two *L. obscura* lineages were nearby species by their morphology, as well as *Lobophora lamourouxii* and *L. thailandensis* sp. nov., which were also contiguous species. In contrast, *L. chumphonensis* sp. nov. distinguished itself from both lineages by its morphology. Hierarchical cluster analysis of morphological traits (thallus thickness, height, width, and morphology) revealed two clear groupings of species in different colors (Figure 3). Thallus height (R^2^ = 0.99, *p* < 0.001) and growth forms (R^2^ = 0.36, *p* < 0.05) contributed to the clustering of Lobophora species, whereas thallus thickness (R^2^ = 0.09, *p* > 0.05) and thallus width (R^2^ = 0.09, *p* > 0.05) did not significantly. In the study areas, one parrotfish (*Scarus rivulatus* Valenciennes, 1840) and three rabbitfishes (*Siganus canaliculatus* Park, 1797, *S. guttatus* Bloch, 1787, and *S. virgatus* Valenciennes, 1835) were recorded. *Lobophora lamourouxii* and *L. obscura*12 were found only on Taen Island at depths of approximately 3.5 m, with low herbivory pressure, and *L. chumphonensis* sp. nov. was found in only three locations at depths of >3 m, with moderate herbivory pressure. On the other hand, *L. obscura*13 and *L. thailandensis* sp. nov. are both common in the western Gulf of Thailand and can be found at depths of 1–7 m with moderate herbivory pressure.

**Table 1 plants-11-03349-t001:** Comparison of morphological and anatomical characteristics among species of the brown alga *Lobophora* from the western Gulf of Thailand.

Characteristics	Algal Species					CV (%)
	*L. chumphonensis* sp. nov.	*L. lamourouxii*	*L. obscura*12	*L. obscura*13	*L. thailandensis* sp. nov.	
**Thickness**						36.57
Average	90.3 ± 9.8 ^b^	95.1 ± 9.8 ^b^	190.8 ± 28.5 ^a^	177.0 ± 13.0 ^ab^	98.4 ± 9.1 ^b^	
Min–Max	75–105	86–113	162–240	156–191	87–110	
**Number of cells**						32.06
Average	8	5	10	8	8	
Min–Max	5–11	5	8–12	5–11	4–12	
**Number of dorsal cells**						42.30
Average	4	2	5	4	4	
Min–Max	2–6	2	4–6	2–6	1–7	
**Number of ventral cells**						33.33
Average	3	2	4	3	3	
Min–Max	2–4	2	3–5	2–4	2–4	
**Medulla length**						15.22
Average	78.9 ± 7.3 ^a^	81.3 ± 4.4 ^a^	65.7 ± 16.7 ^a^	71.1 ± 8.0 ^a^	71.7 ± 5.6 ^a^	
Min–Max	71–90	75–87	36–82.5	59–83	63–78	
**Medulla height**						41.68
Average	21.3 ± 3.5 ^ab^	36 ± 4.14 ^a^	20.4 ± 2.9 ^ab^	13.5 ± 1.3 ^b^	13.8 ± 1.7 ^b^	
Min–Max	21–25.5	30–42	18–24	12–15	12–17	
**Medulla width**						27.58
Average	27.9 ± 6.9 ^a^	17.4 ± 2.8 ^a^	26.7 ± 5.6 ^a^	27.9 ± 4.3 ^a^	19.8 ± 2.0 ^a^	
Min–Max	21–39	14–21	20–35	23–35	17–23	
**Dorsal height**						60.30
Average	28.8 ± 4.3 ^abc^	22.2 ± 1.4 ^bc^	67.5 ± 6.7 ^a^	64.8 ± 7.0 ^ab^	12.6 ± 2.2 ^c^	
Min–Max	24–36	21–24	60–75	40.5–54	9–15	
**Ventral height**						41.75
Average	24.3 ± 3.5 ^b^	21.0 ± 4.2 ^b^	50.4 ± 6.1 ^a^	36.9 ± 5.9 ^ab^	20.7 ± 3.6 ^b^	
Min–Max	18–31.5	15–27	42–60	9–15	18–27	
**Sporangium height**		n/a		n/a	n/a	n/a
Average	44.7 ± 5.6		59.4 ± 4.0			
Min–Max	36–48		54–66			
**Thallus**						
Growth forms	Crustose	Decumbent	Decumbent	Prostrate, Crustose	Prostrate, Decumbent, Crustose	
Colors	Dark brown	Yellow–brown	Light brown	Greenish to light brown with grey band	Dark brown to orange brown	

n/a: data not available. ^a–c^: means in the same row with different superscript letters are significantly different (*p* < 0.05). CV: coefficient of variation value for each characteristic.

### 2.3. Interactions between Lobophora and Corals

*Lobophora*–coral interactions in this study focused on *Lobophora* growing on, at the base, or at the vicinity of live corals. Three species of *Lobophora* (*L. chumphonensis* sp. nov., *L. obscura*13, and *L. thailandensis* sp. nov.) were in direct physical contact with live corals, whereas *L. obcura*12 and *L. lamourouxii* were not found near corals (Appendix A). *Lobophora chumphonensis* sp. nov. interacted only with colonies of the massive coral *Porites*, by growing on bare space between *Porites* colonies. Signs of pink line syndrome were systematically observed on the edge *L. chumphonensis* sp. nov. thalli attached to *Porites* lutea (Appendix A). *Lobophora obscura*13 was the second most important species found in close interaction with the massive coral *Porites*. Some individuals of *L. obscura*13 were found underneath the branching coral *Pocillopora acuta*. Moreover, pink line syndrome was occurred, minor bleaching of *Porites lutea* was observed, and overgrowth by *L. obscura*13 with crustose habit covered unhealthy *Favites* colonies at Kula Island (Appendix A–d). *Lobophora thailandensis* sp. nov. mostly interacted with the massive coral *Porites*, and occurrences of pink line syndrome was documented as well (Appendix A).

## 3. Discussion

This research is the first taxonomic revision of the brown alga genus *Lobophora* using both morphological and molecular data in the Gulf of Thailand and reports of *Lobophora* effects on scleractinian corals. Moreover, two new species are officially described as *L. chumphonensis* sp. nov. and *L. thailandensis* sp. nov. 

*Lobophora* species are typically found in association with corals in the western Gulf of Thailand, particularly on Tao and Taen Islands [35,36]. *Lobophora* were mostly found at depths of less than 3 m in the western Gulf of Thailand, which is consistent with our findings. *Lobophora lamourouxii* is a new species record in Thailand, and it was recently discovered in Singaporean waters [37,38].

Our results revealed that *Lobophora* species mostly interact with the massive coral *Porites lutea*. *Porites* is susceptible to macroalgae competition, with wide reports of coral tissue damages and bleaching when adjacent to *Lobophora* or directly in contact with *Lobophora* [11,16,18]. 

Diaz-Pulido and McCook [39] demonstrated that *Lobophora* propagules could not settle on healthy tissue of coral species, but only on bare skeleton next to healthy tissue. In this study, we discovered that *L. chumphonensis* sp. nov. appears to settle on bare skeleton between *Porites lutea* colonies. According to Puk et al. [40], *Lobophora* sp.82, which is related to *L. chumphonensis* sp. nov., grows at the base of *Porites cylindrica* colonies, and Jompa and McCook [11] concluded that the coral *P. cylindrica* provides a refuge for *Lobophora* from herbivory.

Interestingly, our results show that *L. obscura*13 with an encrusting form grows under the branching coral *Pocillopora acuta*, which is the first report of this effect with the branching coral genus *Pocillopora*. Some *Lobophora* species grow close to corals, under and among coral branches, where they are protected from herbivores [22,41].

High percentages of algal cover have been reported in Kula Island coral reefs [42], consisting of *L. obscura*13 and *L. thailandensis* sp. nov., which both occupy large areas and compete mutually for space. Overall, *Lobophora* species do not represent a specific threat to corals when the coverage of macroalgae is controlled by coral defenses and herbivores [8], with only one case of aggressive coral overgrowth documented in healthy coral reefs by the species *L. hederacea* on the coral *Seriatopora caliendrum* [10]. However, *Lobophora* overgrowth is most likely caused by a severe event, such as a storm, thermal stress, or mass bleaching [8]. The high sediment load from the mainland may have caused the overgrowth of *Lobophora* on Kula Island, as the water was relatively turbid [43]. Furthermore, because the depth of the study sites is very shallow, studies suggest that the abundance of herbivorous fishes is inversely related to water clarity. As a result, declining water quality can negatively affect grazing pressure on algal communities in coastal areas. [44,45].

In the Gulf of Thailand, Müller et al.’s [37] results suggest that the rabbitfish and parrotfish species observed in the Gulf of Thailand avoided feeding on *Lobophora* spp.; however, these observations come from an experimental set-up in a narrow study area, and *Lobophora* can produce a chemical defense to protect themselves against herbivory [22]. The importance of grazing on *Lobophora* in natural environments and on a broader scale remains undocumented. In addition, other publications reported active grazing from *Scarus* spp. and *Siganus* spp. in adults and recruit of *Lobophora* [40,46], indicating that in natural conditions species from both genera might control the growth and spread of this alga in Thailand’s reefs. Previously, contrasting reports were made on the relationship between *Lobophora* spp. growth forms, production of secondary metabolites with defensive roles, and their susceptibility to herbivory [10,47,48]; with [22] pointing toward the primary role of ecological habit (e.g., associational defense and spatial refug-es) as defense strategies against herbivory over chemical or morphological defenses [22].

Our results provide baseline data for the *Lobophora* species diversity and ecology in the western Gulf of Thailand that can be useful for coral management strategies. Further studies are needed to understand the diversity and distribution of *Lobophora*, including their interaction in other parts of the Gulf of Thailand. The allelopathic effects of *L. obscura*13 and *L. thailandensis* sp. nov. and bioactive substances from *Lobophora* species in this study will be further studied.

## 4. Materials and Methods

### 4.1. Sampling Sites

A total of 36 *Lobophora* specimens were collected from 15 sites within coral reefs at Chumphon and Ang Thong Archipelagoes, Prachuap Khiri Khan and Samui islands, the western Gulf of Thailand, from January to March in 2020 (Appendix A). Details of algal specimens are provided in Appendix A. Sampling was carried out at 1–7 m depth by snorkeling or SCUBA diving. Algal specimen fragments were preserved in silica gel for molecular analyses, and remains of the specimens were processed as herbarium specimens, later deposited at the Natural History Museum at Klong Ha, Khlong Luang District, Pathum Thani Province, for morphological and anatomical analyses.

### 4.2. DNA Extraction and Phylogenetic Analyses

The DNeasy Plant minikit (Qiagen, Hiden, Germany) was used to extract algal specimens’ genomic DNA, following the manufacturer’s instruction. Two chloroplast genes (*rbc*L, *psb*A) and one mitochondrial gene (*cox*3) were selected for molecular analyses due to their importance in recent *Lobophora* phylogenetic studies, e.g., [2,4,7,49]. Information on primers, PCR, and sequencing conditions are listed in Appendix A. Phylogenetic trees were reconstructed using a maximum likelihood approach in PhyML v.3.0 [50] based on individual and concatenated alignments of the *cox*3 (724 bp), *psb*A (1030 bp), and *rbc*L (1348 bp) sequences. Genbank accession data used in this study are listed in Appendix A.

### 4.3. Morphological and Anatomical Characteristics

Morphological analyses consisted of in situ observations of the thallus color and growth forms of *Lobophora* specimens following [2]. Dried thalli from the herbarium were rehydrated in seawater, and both transverse and longitudinal sections of the middle part of the plant were made manually using a razor blade and then mounted on a glass slide for microscopic observations. The number and size of both dorsal and ventral cortical cells, including the size of the medulla layer, were measured following [2]. 

### 4.4. Ecological Studies

Ecological analyses consisted in the identification of the main substratum, categorized as follows: (1) growing on live corals, (2) growing at the base of live coral or vicinity of live coral including dead coral, (3) coral rubbles and bedrock, and (4) growing on seaweed bed. Interactions between *Lobophora* species and corals were observed using three permanent belt transects at each study site; the effects of *Lobophora* on corals were conducted from the photo belt transects. Photo belt transects were used as described by [51], consisting of a 30 m transect laid across reef-dominated systems on each site. Photographs were directly taken within 50 cm on each side of belt transect, using a digital camera (Olympus Tough TG-6). Coral species were identified with the genus levels according to [52]. Both negative and positive effects of *Lobophora* towards corals in the western Gulf of Thailand were recorded. The number of thallus of each species were recorded from the photo belt transect and categorized into three mains: (1) growing on live coral, (2) growing at vicinity of live coral, (3) non-coral substrates to analyze quantification of interaction between *Lobophora* and corals and expressed in percentage (100% for each *Lobophora* species). In addition, fish species were identified and counted in situ, and herbivory pressure was estimated. An underwater visual census consisted of three 30 m long and 2 m wide transect per site, performed by Felipe M.G. Mattos (Academia Sinica, Taiwan). Herbivore pressure was categorized from the average abundance of herbivorous fish from where *Lobophora* species were found as follows: (1) low pressure: 0–50 ind./100 m^2^, (2) moderate pressure: 51–100 ind./100 m^2^, and (3) high pressure: >100 ind./100 m^2^. In addition, reef types and depths where samples were collected were recorded.

### 4.5. Statistical Analyses

To compare the similarities between the characteristics of algal species, Tukey’s HSD was used to perform from the package “agricolae” in R. Finally, coefficients of variation for each characteristic were calculated in R to compare results for particular characteristics.

Similarities among species were analyzed in R by using morphological traits (growth forms, thallus thickness, thallus height, and thallus width) as predictors to test for morphological differences among species and create clusters based on algal characteristics. A distance matrix was built by the Euclidean method in the R package “cluster” (Appendix A) to examine the distance between species and later build a hierarchical cluster. The linear relationships between two predictors *Lobophora* species were also analyzed (Appendix A). A hierarchical cluster was built with “hclust” in the R package “cluster” with “ward.D2” agglomeration to analyze the similarities between the morphological traits in *Lobophora* species [53]. 

## Figures and Tables

**Figure 1 plants-11-03349-f001:**
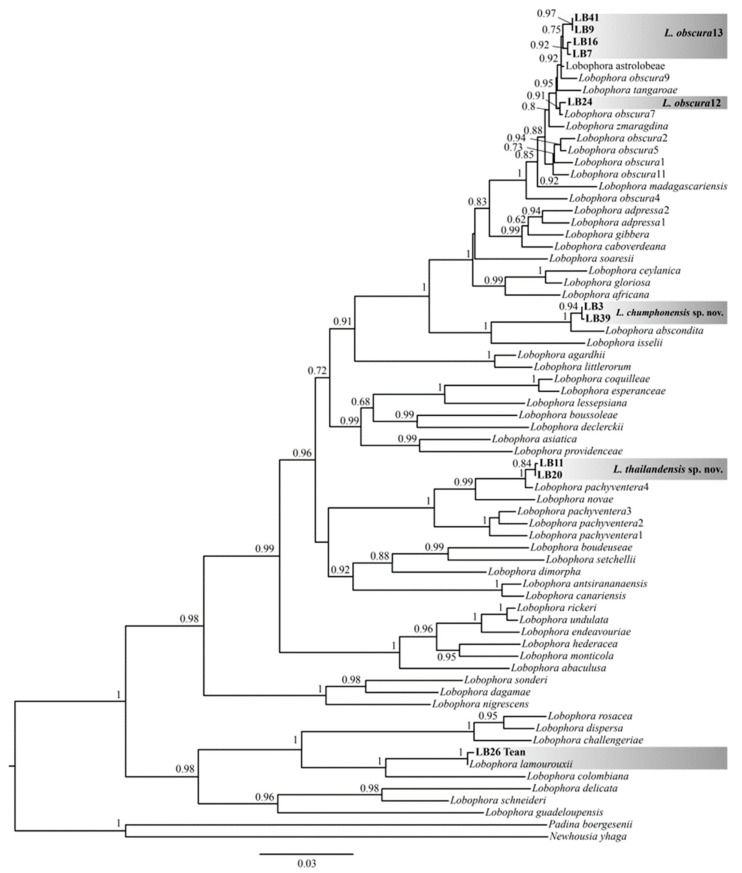
Species-level maximum likelihood (PhyML) phylogenetic tree of the brown algal genus *Lobophora* (Dictyotlaes, Phaeophyceae) constructed using one mitochondrial (*cox*3) and chloroplast (*psb*A and *rbc*L) sequences (3102 bp). Branch support values (aLRT) are displayed in the nodes.

**Figure 3 plants-11-03349-f003:**
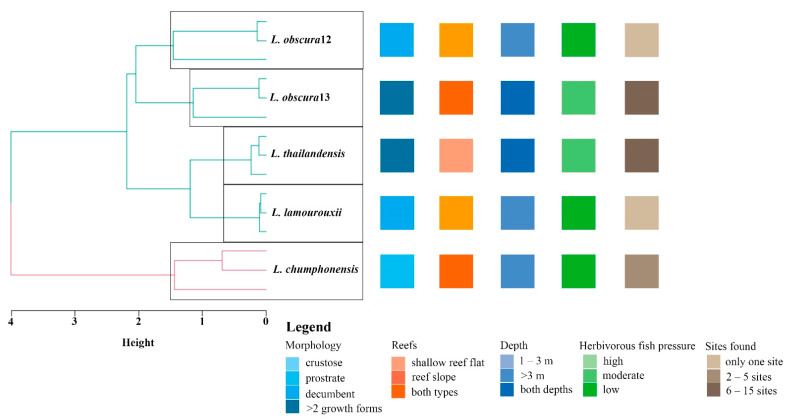
Hierarchical cluster analysis of *Lobophora* species by using morphology, thallus thickness, height, length, and width as predictors. Reef types, depth, herbivorous fish pressure, and sites found were not used to analyze hierarchical cluster. Reefs and depth presented where the *Lobophora* can be found. Herbivorous fish pressure categorized by the abundance of herbivorous fish occurs in areas. The number of sites found is shown.

## Data Availability

The partial genomic sequences used in this study will be available on October 31st at The National Center for Biotechnology Information (NCBI) database, or by prior request from the first author. The raw data used in this study are available upon request from the first author.

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
