# Peer review of "Diversity and Ecology of Lobophora Species Associated with Coral Reef Systems in the Western Gulf of Thailand, including the Description of Two New Species"

_plants, 2022, doi:10.3390/plants11233349_

Round 1
Reviewer 1 Report
The reviewed paper dedicated to study the biodiversity and ecology of the Lobophora species from the western Gulf of Thailand with the description of several new species. This data is very interesting and important for phycology. In the study, the morphological, molecular-genetic and ecological methods were used. Conclusions are supported by results and discussion. The manuscript is well illustrated and clear.
I can recommend the MS for publication in journal “Plants” after correction some mistakes.
Lines 12-30 and further in the MS: Latin names should be italicized.
Line 23: Remove the extra dot after the words “sp. nov.”
Lines 61-62: I think, that you can delete the numbers 1 and 2 from this sentence.
Line 67: At the beginning of the paragraph an indent is required.
I do not understand why you are describing new species in Table 2. It is better to do this in the main text of the paper.
Lines 262-270, 273-282: Delete this information.
Author Response
Dear Reviewer1
I already improved my manuscript following your comments.
Your sincerely,

Reviewer 2 Report
Manuscript "Diversity and ecology of Lobophora species associated to coral reef systems in the western Gulf of Thailand" is very interesting.
General comments:
Authors investigated the diversity of Lobophora in the western Gulf of Thailand using morphological and molecular data, as well as their interactions with scleractinian corals.
Authors analysed 36 Lobophora specimens from 15 sites in the western Gulf of Thailand. They used molecular and morphological observations.
Description of Statistical analyses is very poor and needs correction. Lack information about distribution of observed traits. "hierarchical cluster analysis": What method? ANOVA? Interactions? Relationships?
Detailed comments:
The brown macroalgal genus Lobophora plays important ecological roles in many marine ecosystems. This group received much attention over the past decade, and a considerable number of new species were identified globally.
Introduction is perfect.
My suggestion:
Line 12: Lobophora - italic.
Lines 22-24: "L. chumphonensis", "L. thailandensis", "L. lamourouxii", "L. obscura" and "L. obscura" - italic.
Table 1 needs LSD or HSD values and homogeneous groups.
Table 1 needs coefficients of variation for particular characteristics.
Lines 123-137 "2.3. Interactions between Lobophora and corals": How analysed?
Lines 238-241: description of Statistical analyses is very poor and needs correction. Lack information about distribution of observed traits. "hierarchical cluster analysis": What method? ANOVA? Interactions? Relationships?
Paper needs major revision.
Author Response
Dear Reviewer2
Thank you for your suggestion esspecially for statistic description. We improved some writing and data as your recommendation see in Response to Reviewer file. Moreover, we added some supplymentary files to support the data in revised manuscript.
Your sincerely,
Anirut Klomjit

Round 2
Reviewer 2 Report
Now, all is ok.